# Postoperative complications and hospital costs following small bowel resection surgery

**Dong-Kyu Lee[1], Ashlee Frye[2], Maleck Louis[2], Anoop Ninan Koshy[3], Shervin Tosif[2], Matthew Yii[2], Ronald Ma[4], Mehrdad Nikfarjam[5], Marcos Vinicius Perini[5], Rinaldo Bellomo[6,7], Laurence Weinberg[2,5]***

**1** Department of Anesthesiology and Pain Medicine, Korea University Guro Hospital, Seoul, Republic of Korea, **2** Department of Anesthesia, Austin Health, Heidelberg, Australia, **3** Department of Cardiology, Austin Health, Heidelberg, Australia, **4** Business Intelligence Unit, Austin Health, Heidelberg, Australia, **5** Department of Surgery, The University of Melbourne, Austin Health, Heidelberg, Australia, **6** Department of Intensive Care, Austin Health, Heidelberg, Australia, **7** Data Analytics Research & Evaluation (DARE) Centre, Austin Hospital and The University of Melbourne, Melbourne, Victoria, Australia

* laurence.weinberg@austin.org.au

**Data Availability Statement:** All relevant data are within the manuscript and its Supporting Information files.

**Funding:** The author(s) received no specific funding for this work.

## Abstract

### Background

Postoperative complications after major gastrointestinal surgery are a major contributor to hospital costs. Thus, reducing postoperative complications is a key target for cost-containment strategies. We aimed to evaluate the relationship between postoperative complications and hospital costs following small bowel resection.

### Methods

Postoperative complications were recorded for 284 adult patients undergoing major small bowel resection surgery between January 2013 and June 2018. Complications were defined and graded according to the Clavien–Dindo classification system. In-hospital cost of index admission was calculated using an activity-based costing methodology; it was reported in US dollars at 2019 rates. Regression modeling was used to investigate the relationships among a priori selected perioperative variables, complications, and costs.

### Findings

The overall complication prevalence was 81.6% (95% CI: 85.7–77.5). Most complications (69%) were minor, but 22.9% of patients developed a severe complication (Clavien–Dindo grades III or IV). The unadjusted median total hospital cost for patients with any complication was 70% higher than patients without complications (median [IQR] USD 19,659.64 [13,545.81–35,407.14] vs. 11,551.88 [8,849.46–15,329.87], P < 0.001). The development of 1, 2, 3, and ≥ 4 complications increased hospital costs by 11%, 41%, 50%, and 195%, respectively. Similarly, more severe complications incurred higher hospital costs (P < 0.001). After adjustments were made (for the Charlson Comorbidity Index, anemia, surgical urgency and technique, intraoperative fluid administration, blood transfusion, and hospital readmissions), a greater number and increased severity of complications were associated

**Competing interests:** The authors have declared that no competing interests exist.

**Abbreviations:** ASA, American Society of Anesthesiologist; CCI, Charlson Comorbidity Index; CI, Confidence Interval; CVD, Clavien Dindo Classification; eGFR, estimated glomerular filtration rate; ICU, Intensive Care Unit; ICD, International Statistical Classification of Diseases; NPC, No postoperative complication; PC, Postoperative complication; POD, postoperative day; RBC, Red blood cell; SD, standard deviation; USD, United Stated Dollar.

with a higher adjusted median hospital cost. Patients who experienced complications had an adjusted additional median cost of USD 4,187.10 (95% CI: 1,264.89–7,109.31, P = 0.005) compared to those without complications.

## Conclusions

Postoperative complications are a key target for cost-containment strategies. Our findings demonstrate a high prevalence of postoperative complications following small bowel resection surgery and quantify their associated increase in hospital costs.

## Trial registration

Australian Clinical Trials Registration number: 12620000322932

## Introduction

Identifying the major cost drivers associated with surgical procedures can enable clinicians and hospital administrators to make more informed decisions regarding resource allocation. In turn, these decisions can help guide cost-reduction strategies. Although some studies have evaluated the detailed costs of complications associated with abdominal surgery [1–4], few studies have explored the health costs of complications following small bowel resection surgery. The incidence of small intestine cancer is increasing. In the past 10 years, rates of new small intestine cancer have been rising on average 2.3% each year in the United States of America [5]. In the United Kingdom, the incidence rate has increased by 56% in the last 10 years [6]. In Australia, a similar trend is seen with the age-standardized incidence rate increasing from 0.7 cases per 100,000 persons to 2.2 per 100,000 between 1982 and 2018 [7]. As international healthcare costs are significant and increasing, concerns about government expenditure have been growing. This has resulted in the threat of rapidly increasing surgical costs being moved to the forefront of public debate.

To address the cost of complications in patients undergoing small bowel resection, we conducted a retrospective cost analysis study. The primary aim of this study was to evaluate the relationship between postoperative complications and hospital costs. The secondary aim was to identify the major cost drivers associated with complications. We hypothesized that increasing complication severity and count would be associated with increased hospital costs.

## Materials and methods

### Study design

The Austin Health Human Research Ethics Committee approved this study and provided a waiver for participant consent (Audit/19/Austin/88). The study protocol was registered in the Australian–New Zealand Clinical Trials Registry (No:12620000322932), which is accessible from http://www.anzctr.org.au/Trial/Registration/TrialReview.aspx?ACTRN= 12620000322932.

This study is reported following the Strengthening the Reporting of Cohort Studies in Surgery (STROCSS) guidelines [8]. We conducted a single-center cohort study with retrospective data collection to determine complications and costs associated with postoperative complications following small bowel resection surgery.

## Setting

This study was conducted at a public university teaching hospital in Australia, which has a high volume of abdominal surgery. Inclusion criteria included adult patients (age > 18 years) who had undergone primary elective or emergency small bowel resections between January 2013 and June 2018. Patients were identified using the International Statistical Classification of Diseases and Related Health Problems 10th Revision (ICD-10) and codes specific to small bowel resection (S1 Table in S1 Appendix). We included all open and laparoscopic surgical techniques in the sample.

We excluded patients who had undergone primary colonic, rectal, or anal resection. We also excluded those who had small bowel enterectomy for an isolated stoma reversal and small bowel resection that was secondary to another major procedure (e.g., small bowel resection for gut ischemic post-cardiac surgery).

## Outcomes

Total hospital cost was defined as the sum of direct and indirect in-hospital costs of index admission for small bowel resection surgery. Raw costing data were obtained from our institution's clinical informatics and costing center, which included patient-care activities relating to anesthesia, operative theater, the intensive care unit (ICU), ward, medical consults, allied health, pathology, blood products, pharmacy, radiology, medical emergency team calls, and hospital-in-the-home. Costs incurred during the preoperative period were excluded from data analysis to prevent potential confounding due to preoperative cost drivers. In-hospital costs arising from any unplanned readmissions within 30 days of discharge were added to the total cost. Costs were inflated to 31 December 2019 based on the end-of-fiscal-quarter Australian Consumer Price Index [9]. The costs were then converted to United States dollars (USD) based on the market rate on 31 December 2019 [10]. In-hospital costs were calculated according to an activity-based costing methodology that allocated costs based on service volume. Postoperative complications during index admission were coded by the Data Analytics Research and Evaluation Center and were independently cross-checked with a complete chart review by two authors.

## Definitions

Postoperative complications were defined as being any deviation from the normal postoperative course during the index admission for small bowel resection surgery, which was guided by the European Perioperative Clinical Outcome definitions [11]. The severity of complications was graded according to the Clavien–Dindo (CVD) system [12], which is a validated classification system that categorizes complication severity based on the level of required treatment. CVD grade I includes any deviation from the normal postoperative course that does not require intervention—excluding antiemetics, antipyretics, analgesia, diuretics, electrolytes, and physiotherapy. CVD grade II requires pharmacological treatment, blood transfusion, or total parenteral nutrition. CVD grade III requires radiological, surgical, or endoscopic intervention. CVD grade IV includes any life-threatening complications that require intensive care management, and CVD grade V is when death occurs. Patients were stratified into groups based on the worst complication severity recorded.

The length of stay was defined as the number of days from completion of surgery to discharge, excluding days on leave or in the hospital-in-the-home unit. Readmissions were defined as any unplanned readmission 30 days post-discharge. Mortality was defined as inpatient mortality according to the definition of CVD grade V classification (i.e., death of a patient).

## Data sources

Data collection was undertaken using Cerner® electronic health records (Cerner Millennium, Kansas USA), which contained prospectively recorded perioperative and patient health variables. The collected perioperative data included patient demographics, body mass index, history of smoking within one year of surgery, history of alcohol abuse, the American Society of Anesthesiologists score [13], the Charlson Comorbidity Index (CCI) [14], diagnosis of malignancy, preoperative chemotherapy within three months, history of previous small or large bowel resection, and history of previous abdominal surgery. The collected preoperative laboratory data included hemoglobin concentration, platelet concentration, serum albumin and bilirubin concentrations, serum creatinine level, and estimated glomerular filtration rate (eGFR).

Collected intraoperative data included operation urgency (i.e., emergency or elective), surgical techniques (e.g., open laparotomy or laparoscopic surgery), operative time, intraoperative use of vasoactive medications (e.g., inotropes or vasopressors) and administered crystalloid, and colloid volumes. The collected postoperative data included ICU admission, ICU care duration, length of hospital stay, destination following discharge, 30-day readmission, and postoperative hemoglobin concentration. Data related to blood product transfusions were collected, which included preoperative, intraoperative, and postoperative allogeneic red blood cell transfusion.

## Statistical methods

Statistical analysis was performed using IBM SPSS Statistics for Windows, version 23 (IBM Corp, Armonk, NY, USA) and R version 4.0.0 (R Development Core Team, Vienna, Austria, 2020). Study patients were classified into two groups: the 'no complication group' (NPC) for patients who did not experience postoperative complications and the 'complication group' (PC) for patients who experienced one or more postoperative complications.

Before statistical analysis, missing data analysis was performed to detect more than 5% missing values for all variables. For variables with less than 5% of missing values, statistical analysis excluding cases by analysis was planned. The multiple imputation method was performed in cases of missing values of more than 5%. All continuous variables were tested for normality using the Q-Q plot. When the normality assumption was violated, non-parametric statistical methods were considered. Comparative statistics were estimated using Student's t-test, Mann–Whitney U test, Chi-square test, Cochran–Armitage test, and Fisher's exact test, depending on the characteristics of the variables and the results of the required assumption tests. Data are presented as mean ± standard deviation (SD) or median [IQR] for continuous variables and number (percentile) for categorical variables. Comparative results are presented with a P-value and corresponding effect size. A two-tailed P-value below 0.050 was considered to be statistically significant.

Total hospital costs in relation to complications were analyzed using unadjusted and adjusted hospital costs. For the adjusted analyses, costs were analyzed according to the occurrence, number, and severity of complications using covariates of both clinical and statistical importance. To evaluate the unadjusted relationship between postoperative complications and hospital costs, the Mann–Whitney U test and the Kruskal–Wallis H test were used. When the Kruskal–Wallis H test revealed significant differences, all multiple pairwise comparisons were performed under a Bonferroni adjusted P-value. Detailed cost items were compared by the groups and the number and severity of complications using the Mann–Whitney U and the Kruskal–Wallis H tests.

For the adjusted hospital cost analysis as the primary outcome, we estimated a bootstrapped quantile regression model. The independent variable was the presence, number, and severity

of complications; the dependent variable was hospital cost. The a priori selected covariates were the CCI, preoperative anemia, emergency surgery, surgical technique, the volume of intraoperative fluid administration, and transfusion during admission. Because hospital cost had a severely positively skewed distribution (skewness of 6.45: 95% CI: 6.20–6.71), we used quantile regression modeling to investigate the cost-driving effects of complications according to low (25th quantile), median (50th quantile), and high (75th quantile) cost brackets. Spearman's correlation analysis was performed to clarify which variables were in a relationship with complications and hospital costs. Based on the correlation analysis results (see S1 Fig in S1 Appendix) and considering the clinical relevance, several variables were then selected for the adjusted regression analysis. Bootstrapped quantile regression was performed using the 'quantreg' package in R [15]. For each estimation, three quantile regression models were included: the 25th percentile, the 50th percentile (median), and the 75th percentile. The estimated models were evaluated using pseudo-$R^2$ and the Akaike information criterion. The Wald test was used to compare the estimated parameters from different percentile or linear regressions. Heteroskedasticity was evaluated using the studentized Breusch–Pagan test. To assess multicollinearity, variance inflation factors were used. The estimated values are expressed with a 95% confidence interval (95% CI). To correct for multiple comparisons, the Bonferroni correction was applied.

## Results

### Baseline patient characteristics

From 383 potentially eligible patients who had undergone small bowel resection at our institution, 35 (9.1%) were excluded. The reasons for exclusion were age less than 18 years (n = 2), small bowel resection aborted (n = 7), and small bowel resection secondary to another major surgical procedure (n = 26). Thus, 348 patients were included in the final statistical analysis.

Among the data of 348 patients, missing data analysis demonstrated fewer than 5% missing values for all variables. The variables with the highest missing data rate were 'preoperative bilirubin concentration' (4.3%), 'intraoperative crystalloid administration volume' (3.4%), 'preoperative albumin concentration' (2.9%), and 'lowest postoperative hemoglobin concentration' (0.9%). Statistical analysis was performed as a complete case analysis. Patients in the PC group were older, had greater comorbidities (as reflected by the CCI), and had a higher American Society of Anesthesiologists (ASA) physical status classification compared to NPC patients (all P < 0.001). Further, patients in the PC group were more likely to be anemic (P = 0.034) and to have reduced renal function (P < 0.001). The proportion of patients with hypoalbuminemia was also higher in the PC group than the NPC group (55.2% vs. 34.4%, respectively, P = 0.004). The baseline characteristics and preoperative variables of patients are presented in Table 1.

### Complications

Overall, 284 patients experienced one or more postoperative complications (PC group incidence rate of 81.6% [95% CI: 85.7–77.5]), with 64 patients discharged without complications (NPC group). The numbers of patients with 1, 2, 3, and ≥ 4 complications were 59 (20.8%), 49 (17.2%), 56 (19.7%), and 120 (42.3%), respectively. The median [IQR] number of complications per patient in the PC group was 3 [2–5]. Of the PC patients, 59 (20.8%) were CVD grade I, 137 (48.2%) CVD grade II, 21 (7.4%) CVD grade III, and 44 (15.5%) CVD grade IV. The mortality rate was 8.1% (23 deaths). The correlation coefficient between the number and severity of complications was 0.709 (P < 0.001), indicating a high relationship. Of the patients, 86% who developed a grade IV complication and 81% who developed a grade III complication experienced > 4 complications in total. Of the patients who developed a grade II complication,

**Table 1. Preoperative baseline patient characteristics.**

| Variables | | NPC group (N = 64) | PC group (N = 284) | Mean difference | P value | Effect size |
|---|---|---|---|---|---|---|
| Female gender | | 28 (43.8) | 137 (48.2) | - | 0.580 | 0.03 |
| Age (years) | | 53.7 ± 16.9 | 64.8 ± 18.1 | 11.1 (6.2–16.0) | <0.001* | 0.62 |
| BMI (kg/m$^2$) | | 27.02 ± 6.30 | 26.00± 6.35 | 1.06 (-2.87–0.67) | 0.230 | 0.17 |
| Smoker within 1 year | | 13 (20.3) | 47 (16.5) | - | 0.583 | 0.04 |
| Alcohol abuse | | 1 (1.6) | 13 (4.6) | - | 0.481 | 0.06 |
| Malignancy | | 8 (12.5) | 50 (17.6) | - | 0.360 | 0.05 |
| Preoperative chemotherapy | | 2 (3.1) | 13 (4.6) | - | >0.999 | 0.03 |
| Previous small bowel resection | | 14 (21.9) | 78 (27.5) | - | 0.434 | 0.05 |
| Previous abdominal surgery | | 30 (46.9) | 165 (58.1) | - | 0.125 | 0.09 |
| ASA | I | 11 (17.2) | 10 (3.5) | - | <0.001* | 0.37 |
| | II | 36 (56.3) | 79 (27.8) | | | |
| | III | 16 (25) | 124 (43.7) | | | |
| | IV | 1 (1.6) | 68 (23.9) | | | |
| | V | 0 (0.0) | 3 (1.1) | | | |
| CCI | | 3 (1–4), [0:10] | 4 (2–6), [0:15] | - | <0.001* | 0.23 |
| Preoperative anemia† | | 18 (28.1) | 122 (43.0) | - | 0.034* | 0.12 |
| Preoperative platelet count (× 10$^3$/µℓ) | | 274.3 ± 96.8 | 271.6 ± 116.4 | -2.7 (-33.5–28.1) | 0.861 | 0.02 |
| GFR (ml/min/1.73m$^2$)‡ | Over 90 | 32 (50) | 93 (33) | - | <0.001* | 0.22 |
| | 60–89 | 29 (45.3) | 109 (38.7) | | | |
| | 45–59 | 1 (1.6) | 32 (11.3) | | | |
| | 30–44 | 1 (1.6) | 21 (7.4) | | | |
| | 15–29 | 1 (1.6) | 23 (8.2) | | | |
| | Below 15 | 0 (0) | 4 (1.4) | | | |
| Hypoalbuminemia | | 21 (34.4) | 153 (55.2) | - | 0.004* | 0.16 |
| Hyperbilirubinemia | | 2 (3.3) | 41 (15.1) | - | 0.018* | 0.14 |

NPC group: No postoperative complication group, PC group: Postoperative complication group. Data are presented as number (percentile), mean ± SD or median (interquartile range), [Min:Max]. ASA: American Society of Anesthesiologist physical classification system, CCI: Charlson Comorbidity index, eGFR: estimated glomerular filtration rate. Student's t-test, Mann-Whitney U test, Chi-square or Fisher's exact test were used and described corresponding effect size as Cohen's *d* for Student's t-test, common language effect size *r* for U test, Cramér's *V* for Chi-square and Fisher's exact tests.

*: indicates P<0.05

†: Anemia defined as Hb below 130 g/L for men and 120 g/L for women

‡: according to the Kidney Disease: Improving Global Outcomes 2012 Clinical Practice Guideline for the Evaluation and Management of Chronic Kidney Diseases.

64% experienced ≥ 3 complications, 23.7% had 2 complications, and 49.2% had 1 complication. A detailed overview of the specific type and severity of each postoperative complication is presented in S2 Table in S1 Appendix.

## Adjusted costs

S1 Fig in S1 Appendix presents the Spearman correlation analysis results among complications, total hospital cost, and other collected variables. Among the collected variables, the following were selected as covariates: the CCI, preoperative anemia, surgical technique, emergency surgery, the volume of intraoperative fluid administration, and transfusion during admission. The CCI is a representative variable for preoperative patients' status. Preoperative anemia significantly correlated with complications and hospital costs; it was selected as a covariate because of its important clinical relevance. Although its correlation coefficient was weak, the surgical technique consistently correlated with complications and hospital costs.

Emergency surgery was selected as a covariate because it is not included in the CCI classification. Intraoperative fluid volume had a weak relationship with the number of complications and hospital costs; however, as it can be related to postoperative complications, it was selected as a covariate [16]. Transfusion during admission showed a moderate relationship with the severity of complications and hospital costs, and a weak relationship with the number of complications. As transfusion and anemia are associated with the development of postoperative complications, they were selected as covariates [17, 18]. Although age and operative time correlated with complications and hospital costs, they were excluded because of multicollinearity. Other variables were excluded due to a non-significant correlation or clinical irrelevance.

The adjusted hospital costs were independently increased by the presence of complications, their number, and severity (see Fig 1). In PC patients, the median cost after adjusting for covariates was USD 4,187.10 (95% CI: USD 1,264.89–7,109.31, P = 0.005) greater than NPC patients. For the difference in total adjusted hospital cost between PC and NPC patients, the increased severity of complications was more marked in the 75th centile of total cost than in the 25th or 50th centiles (see Fig 1, S3 Table in S1 Appendix).

The total number of complications was associated with a higher median hospital cost. Although the adjusted total hospital cost for patients with 2–3 complications was not significantly increased compared to patients with 0–1 complication, patients who experienced ≥ 4 complications had significantly higher adjusted additional hospital costs compared to those with fewer complications. The cost-driving effects of ≥ 4 complications were increased in the 75th centile and almost double that of the 25th centile of the total cost (see Fig 1, S4 Table in S1 Appendix).

Patients who suffered CVD grades III and IV complications had a significantly higher adjusted additional hospital cost (P = 0.003 for the 25th centile of hospital cost in CVD grade III, otherwise P < 0.001). The additional adjusted hospital cost also tended to increase with

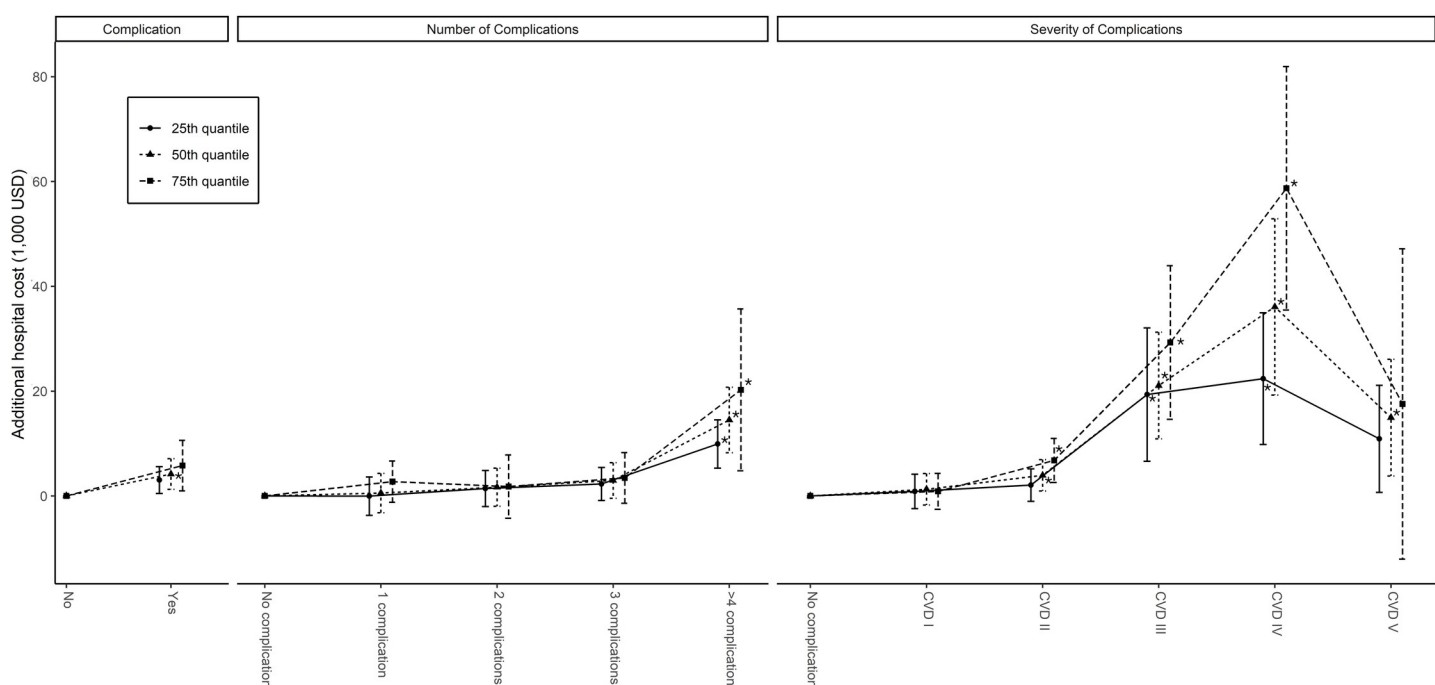

**Fig 1. Complications and adjusted additional hospital cost.** Estimated quantile regression coefficients and 95% confidence intervals for given hospital cost at 25th, 50th, and 75th percentiles. The coefficients of presence, numbers, and severity of complications predicting the driving effects on hospital cost are presented from the left. Severity is measured by the Clavien–Dindo surgical complications classification. *: P < 0.016.

centiles of hospital cost (P < 0.001). The adjusted additive hospital cost in CVD grade II patients was significant in the 50th and 75th percentiles of hospital cost. The adjusted additive hospital cost of CVD grade V patients (in-hospital mortality) had a significant impact on the 50th centile of hospital cost (P = 0.009 for the 50th percentile) but not on the 25th and 75th centiles (see Fig 1, S5 Table in S1 Appendix).

## Unadjusted costs

The unadjusted median total hospital cost of the PC group was 70% higher than the NPC group: median USD 19,659.64 (13,545.81–35,407.14) for the PC group vs. USD 11,551.88 (8,849.46–15,329.87) for the NPC group (P < 0.001) (see Table 2). The development of 1, 2, 3, and ≥ 4 complications increased hospital costs by 11%, 41%, 50%, and 195%, respectively. Similarly, higher grades of complications incurred higher hospital costs (P < 0.001). Compared to NPC patients, those who experienced complications at CVD grades I, II, and III had increased hospital costs of 15%, 60%, and 219%, respectively. The development of a CVD grade IV complication (i.e., organ failure) increased costs by 470%. Patients who died incurred 89% higher costs than NPC patients. The unadjusted median costs according to the number and severity of complications are presented in Table 2 and S2 and S3 Figs in S1 Appendix. A breakdown of the hospital costs according to the cost center where the cost was incurred is presented in S6-S8 Tables in S1 Appendix.

## Intraoperative and postoperative variables

Intraoperative and postoperative variables are presented in Tables 3 and 4. Patients in the PC group were more likely to undergo an open laparotomy and had longer operative times compared to patients in the NPC group. Intraoperatively, patients in the PC group were more

**Table 2. Complications and unadjusted hospital cost.**

| Groups | | Median (IQR), [Range of data] | P value | Effect size |
|---|---|---|---|---|
| NPC group | | 11,551.88 (8,849.46–15,329.87), [5,182.17:27,363.99] | <0.001* | 0.40 |
| PC group | | 19,659.64 (13,545.81–35,407.14), [5,396.18:459,105.19] | | |
| No of surgical complications | One complication | 12,788.65 (9,471.24–17,118.54), [5,396.18:65,176.97] | <0.001* | 0.45 |
| | Two complications | 16,306.07 (13,095.21–20,930.34), [5,566.81:105,855.83] † | | |
| | Three complications | 17,345.77 (13,445.19–22,158.55), [7,807.11:56,318.26] †‡ | | |
| | ≥ Four complications | 34,107.54 (22,414.36–61,458.22), [10,014.17:459,105.19] †‡§¶ | | |
| CVD | Grade I | 13,326.53 (10,540.28–16,902.17), [5,396.18:41,051.24] † | <0.001* | 0.42 |
| | Grade II | 18,558.00 (14,739.34–26,993.63), [6,087.08:105,855.83] † | | |
| | Grade III | 36,814.36 (26,090.30–52,233.58), [16,797.46:99,866.35] †‡§ | | |
| | Grade IV | 65,900.04 (31,882.87–93,717.97), [5,566.81:459,105.19] †‡§ | | |
| | Grade V | 21,815.72 (12,626.47–37,260.71), [7,058.36:92,747.48] †‡** | | |

Hospital cost is presented as USD and a value of inflated to 31 Dec 2019 based on end of fiscal quarter Australian Consumer Price index. Values are presented as median (interquartile range), [Min:Max]. NPC group: No surgical complication group, PC group: Surgical complication group, CVD: Clavien-Dindo surgical complication classification Mann-Whitney U test, Kruskal-Wallis H test were used and described corresponding effect size as common language effect size *r* for Mann-Whitney U test, $\eta_H^2$ (eta squared estimated using H-statistics) for Kruskal-Wallis test.

*: indicates P <0.016. Multiple comparison results

†: P<0.010 vs. NSC group

‡: P<0.010 vs. one complication or P<0.0083 vs. CVD grade I

§: P<0.010 vs. two complications or P<0.0083 vs. CVD grade II

¶: P<0.010 vs. three complications or P<0.0083 vs. CVD grade III

**: P<0.0083 vs. CVD grade IV.

**Table 3. Surgery and postoperative course.**

| Variables | | | NPC group (N = 64) | PC group (N = 284) | Mean difference | P value | Effect size |
|---|---|---|---|---|---|---|---|
| Emergency operation | | | 40 (62.5) | 211 (74.3) | - | 0.065 | 0.10 |
| Technique | Laparotomy | | 33 (51.6) | 213 (75.0) | - | <0.001* | 0.20 |
| | Converted to laparotomy from laparoscopy | | 13 (20.3) | 28 (9.9) | | | |
| | Laparoscopy | | 18 (28.1) | 43 (15.1) | | | |
| Operation time (min) | | | 195.3 ± 63.3 | 221.0 ± 87.1 | 25.7 (7.0–44.4) | 0.007* | 0.31 |
| Intraoperative: patients receiving vasoactive medications | | | 34 (53.1) | 213 (75.0) | - | 0.001* | 0.19 |
| Total intraoperative fluid volume (excluding blood products) | | | 1954.8 ± 898.7 | 2249.9 ± 1383.7 | 295.0 (16.9–573.1) | 0.038* | 0.23 |
| | Crystalloid | Volume (ml) | 1900.8 ± 820.0 | 2031.0 ± 1084.6 | 130.2 (-112–372) | 0.290 | 0.13 |
| | Colloid | No of patients | 3 (4.8) | 59 (21.5) | - | 0.002* | 0.17 |
| | | Volume (ml) | 600 (100–600), [100:1000] | 500 (100–500), [100:2000] | - | 0.661 | 0.06 |
| No of patients requiring ICU admission | | | 4 (6.3) | 128 (45.1) | - | <0.001* | 0.31 |
| ICU admission duration (hours) | | | 16.50 (8.25–48.75), [6:59] | 50.00 (18.00–126.75), [3:106.44] | | 0.110 | 0.14 |
| Length of hospital stay in days | | | 5 (4–6), [2:31] | 11 (7–19), [1:189] | - | <0.001* | 0.44 |
| 30-day readmission: no of patients | | | 8 (12.5) | 42 (14.8) | - | 0.699 | 0.03 |

NPC group: No postoperative complication group, PC group: Postoperative complication group. Data are presented as number (percentile), mean ± SD or median (interquartile range), [Min:Max]. Student's t-test, Mann-Whitney U test, Chi-square or Fisher's exact test were used and described corresponding effect size as Cohen's *d* for Student's t-test, common language effect size *r* for U test, Cramér's *V* for Chi-square and Fisher's exact tests.

*: indicates P <0.05

†: summed volume of 4% and 20% albumin.

likely to require vasoactive medication and receive larger volumes of fluid than the NPC group. Postoperatively, patients in the PC group were more likely to have anemia and a lower hemoglobin concentration than the NPC group. No patient in the NPC group received a blood transfusion during their admission, whereas 25.4% of patients in the PC group received

**Table 4. Postoperative hemoglobin concentration and transfusion.**

| Variables | | NPC group (N = 64) | PC group (N = 284) | Mean difference | P value | Effect size |
|---|---|---|---|---|---|---|
| POD 1: hemoglobin (g/L) | | 123.4 ± 15.7 | 111.1 ± 20.0 | -12.3 (-16.9 – -7.7) | <0.001 | 0.64 |
| POD 1: patients with anemia | | 32 (51.6) | 212 (74.9) | - | <0.001 | 0.2 |
| Lowest hemoglobin: POD 1–7 | | 114.3 ± 16.3 | 96.0 ± 19.1 | -18.3 (-23.5 – -13.2) | <0.001 | 0.98 |
| Patients with anemia: POD 1–7 | | 46 (75.4) | 268 (94.4) | - | <0.001 | 0.25 |
| POD where Hb was lowest | | 3 (2–4), [1:7] | 3 (2–5), [0:7] | - | 0.022 | 0.12 |
| Preoperative transfusion | Patients requiring RBC transfusion | 0 (0.0) | 17 (6.0) | - | 0.050 | 0.11 |
| | Units of RBC‡ | NA† | 2 (1–7), [1:12] | - | NA† | NA† |
| Intraoperative transfusion | Patients requiring RBC transfusion | 0 (0.0) | 24 (8.5) | - | 0.011 | 0.13 |
| | Units of RBC | 0 (0–0), [0:0] | 0 (0–0), [0:20] | - | 0.016 | 0.13 |
| Postoperative transfusion | Patients requiring RBC transfusion | 0 (0.0) | 66 (23.2) | - | <0.001 | 0.23 |
| | Units of RBC‡ | NA† | 2.5 (1–4.25), [1:38] | - | NA† | NA† |
| Transfusion during admission | Patients requiring RBC transfusion | 0 (0.0) | 72 (25.4) | - | <0.001 | 0.24 |
| | Units of RBC‡ | NA† | 3 (2–6), [1:39] | - | NA† | NA† |

NPC group: No postoperative complication group, PC group: Postoperative complication group. Data are presented as number (percentile), mean ± SD or median (interquartile range), [Min:Max]. POD: Postoperative day(s). Student's t-test, Mann-Whitney U test, Chi-square or Fisher's exact test were used and described corresponding effect size as Cohen's *d* for Student's t-test, common language effect size *r* for U test, Cramér's *V* for Chi-square and Fisher's exact tests

‡: number of transfused blood products in patients who received the corresponding transfusion.

an allogeneic red blood cell transfusion, with a median transfusion volume of 3 units (2–6 units). The PC group had a greater ICU admission rate than the NPC group (45.1% vs. 6.3%, P < 0.001). The median length of hospital stay was 11 days (7–19) in the PC group vs. 5 days (4–6) in the NPC group (P < 0.001).

## Discussion

In this retrospective cost analysis of patients undergoing small bowel resection, we found that 4 out of every 5 patients developed a postoperative complication. Most of these complications (69%) were minor (CVD grades I or II), but almost 1 in 4 patients developed a severe compli-cation (CVD grades III or IV). During their index hospital admission, 8% of patients died. The development of a complication increased the median unadjusted hospital costs by 70%. In line with our hypotheses, hospital costs increased significantly as postoperative complication count and severity increased.

We found that hospital costs significantly increased when patients suffered ≥ 4 complica-tions; moreover, higher grades of complications (i.e., CVD grades III and IV) incurred the highest costs. Complications requiring pharmacological treatment or transfusion (i.e., CVD grade II) had an additive effect on hospital cost in the 75th cost centiles. The additive effect of CVD grades II and V were smaller than those of CVD grades III and IV. Minor complications (i.e., CVD grade I) did not have an additive effect on hospital costs.

There is a lack of literature on the relationship between postoperative complications and hospital costs following small bowel resection surgery, limiting the ability for direct compari-son of results. The prevalence of complications in our study is higher than that reported in the limited amount of available literature. A study by Scott et al. [19] estimated the complication rate of emergency small bowel resection surgery at 46.9%, which was significantly lower than what we found in our study (81.6%). Wancata et al. [20] reported complication rates following small bowel resection of 32% and 27% for malignant and benign small bowel obstructions, respectively. This is reflective of significant variations in the definitions and reporting of post-operative complications in the literature, which limits the ability to compare the prevalence of complications across studies. The classifications of complications are inconsistent among the studies and do not reflect the full spectrum of complications. Minor complications are often considered clinically and financially trivial, so they have generally been omitted when report-ing overall complication prevalence following small bowel resection surgery. This is evident in the study undertaken by Scott et al., which only included 13 complication types, excluded minor complications, and did not grade the complications using a pre-validated classification system. In the current study, we utilized a predetermined, exhaustive, and validated classifica-tion system for the capture of both complication type and severity [9, 10].

Our study highlighted the importance of severe postoperative complications (i.e., CVD grades III and IV) as major drivers of hospital costs and suggested that minor postoperative complications (i.e., CVD grade I) do not significantly increase hospital costs. In our study, CVD grade II complications only had an additive effect on hospital costs in the 50th and 75th cost percentiles. Although there are no similar studies on small bowel resection surgery to compare these results, similar studies have been undertaken for other major abdominal surger-ies [2–4]. Our results are similar to what has been found following rectal resection. For exam-ple, Johnston et al. [3] found that minor postoperative complications (i.e., CVD grade I) did not significantly increase hospital costs following rectal resection. However, studies investigat-ing the relationship between postoperative complications and hospital costs following colonic and liver resections demonstrated different results [2, 4]. These studies found a significant association between minor complications and hospital costs.

Major complications are a consistent driver of higher hospital costs across studies [2–4, 21]. Our study further highlighted the importance of targeting complication severity and count. Grades III and IV complications were associated with an exponential increase in costs, particularly in the highest quartile of hospital cost. This highlights that complications requiring procedural intervention or ICU admissions had the greatest cost implications following small bowel resection. Notably, a significant number of patients with minor complications still experienced > 3 complications, suggesting that the cumulative number of minor complications might also contribute significantly to hospital costs. This provides a cost-effective rationale for early recognition of patient deterioration to prevent increases in the number or severity of complications.

Identifying perioperative variables associated with the development of postoperative complications enables risk stratification of patients and the implementation of targeted complication prevention strategies. We have demonstrated an association between postoperative complications following small bowel resection surgery and the CCI, preoperative anemia, hypoalbuminemia, surgical techniques, and the use of intraoperative vasoactive medications. The CCI has been shown to be associated with the development of postoperative complications following colonic resection, and our study has further supported the CCI as an important perioperative predictor of complications following small bowel resections [4].

Further, our study has contributed to the growing body of literature that has identified hypoalbuminemia as a strong predictor of surgical morbidity and mortality. There have been many studies that have established this association following colorectal surgery [4, 22–25]. Our research found that this association also exists for small bowel resection surgery. Finally, we demonstrated an association between the volume of intraoperative fluid administration and the number and severity of complications. Our study adds to the growing body of literature that has demonstrated intraoperative fluid administration as a predictor of postoperative complications following major abdominal surgery [16].

There were several strengths to our study. We have provided a comprehensive analysis regarding the impact of postoperative complications on hospital costs following small bowel resection surgery. We used a standardized and validated method of classifying complication severity [10]. Further, we analyzed the relationship of cost against the number and severity of complications using a detailed and comprehensive cost database. Our study focused on all complications, regardless of type and severity.

Our study, however, has several limitations. First, this is a retrospective study, so there may be a degree of selection and information bias. However, the effects of this bias on our study outcomes are likely to be minimal due to extensive cross-checking of data entered into the electronic medical records used at our institution. Second, our study was completed in a single institution, which may limit its external validity. However, our center has essentially the same operative, anesthesia, and postoperative care protocols as other tertiary centers. Third, the effect of complications on hospital costs could not be evaluated simultaneously using the number and severity of complications. They were highly correlated with each other but produced high levels of multicollinearity when treated concurrently in the regression analysis. Their combined effect was, therefore, indirectly evaluated. Finally, our study did not investigate long-term clinical and economic outcomes following small bowel resection surgery. Further, our cost analysis did not consider community-related costs, which is an area for future research in this field.

## Conclusion

Small bowel study resection surgery was associated with a high prevalence of complications, which were associated with increased hospital costs. Hospital costs significantly increased

when patients suffered $\geq 4$ complications, experienced complications requiring surgical, endoscopic, or radiologic intervention, or had a life-threatening complication (i.e., organ failure). Additionally, the CCI, preoperative albumin, preoperative anemia, surgical technique, the use of intraoperative vasoactive medication, and the volume of intraoperative fluid administration were associated with postoperative complications. Further research is required to identify predictors of postoperative complications and thus enable targeted and cost-effective prevention strategies.

## Supporting information

**S1 Appendix. Supporting figures and tables.**
(DOCX)

**S1 Data.**
(XLSX)

## Author Contributions

**Conceptualization:** Mehrdad Nikfarjam, Marcos Vinicius Perini, Rinaldo Bellomo, Laurence Weinberg.

**Data curation:** Ashlee Frye, Maleck Louis, Anoop Ninan Koshy, Matthew Yii, Ronald Ma, Laurence Weinberg.

**Formal analysis:** Dong-Kyu Lee, Laurence Weinberg.

**Investigation:** Shervin Tosif, Ronald Ma.

**Methodology:** Dong-Kyu Lee, Anoop Ninan Koshy, Shervin Tosif, Mehrdad Nikfarjam, Marcos Vinicius Perini, Rinaldo Bellomo, Laurence Weinberg.

**Project administration:** Laurence Weinberg.

**Software:** Dong-Kyu Lee.

**Supervision:** Maleck Louis, Laurence Weinberg.

**Validation:** Dong-Kyu Lee, Laurence Weinberg.

**Visualization:** Dong-Kyu Lee, Shervin Tosif, Laurence Weinberg.

**Writing – original draft:** Dong-Kyu Lee, Ashlee Frye, Maleck Louis, Anoop Ninan Koshy, Shervin Tosif, Matthew Yii, Ronald Ma, Mehrdad Nikfarjam, Marcos Vinicius Perini, Rinaldo Bellomo, Laurence Weinberg.

**Writing – review & editing:** Dong-Kyu Lee, Ashlee Frye, Anoop Ninan Koshy, Shervin Tosif, Matthew Yii, Ronald Ma, Mehrdad Nikfarjam, Marcos Vinicius Perini, Rinaldo Bellomo, Laurence Weinberg.

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
