## [Decision Letter · Decision Letter 0]

3 Sep 2020

PONE-D-20-23870

Postoperative complications and hospital costs following small bowel resections

PLOS ONE

Dear Dr. Laurence Weinberg.

Thank you for submitting your manuscript to PLOS ONE. After careful consideration, we feel that it has merit but does not fully meet PLOS ONE’s publication criteria as it currently stands. Therefore, we invite you to submit a revised version of the manuscript that addresses the points raised during the review process.

I would appreciate if you pay careful attention to the reviewer's comments in your response.

We look forward to receiving your revised manuscript.

Kind regards,

Ehab Farag, MD FRCA FASA

Academic Editor

PLOS ONE

Journal Requirements:

2. Thank you for providing information in the Ethics Statement about your ethics board approval and their waiver of the requirement for informed patient consent. However, we ask that you also provide this information within your Methods section.

Reviewers' comments:

Reviewer's Responses to Questions

**Comments to the Author**

1. Is the manuscript technically sound, and do the data support the conclusions?

Reviewer #1: No

2. Has the statistical analysis been performed appropriately and rigorously? 

Reviewer #1: No

3. Have the authors made all data underlying the findings in their manuscript fully available?

Reviewer #1: Yes

4. Is the manuscript presented in an intelligible fashion and written in standard English?

Reviewer #1: No

5. Review Comments to the Author

Reviewer #1: I have three major concerns about the study method.

First, for the determination of complication. Authors needs to clarify how the complications were determined and counted, which is crucial to the validity of the study. Especially, authors need to verify that all the complication are postoperative. For example, myocardial infarction and respiratory failure both have counts under Grade I in supplement table 2?

The other concern is about the use of quantile regression model. While authors claim that hospital cost was negatively skewed (left skewed), supplement graphs showed that hospital cost are right skewed. In this case, I would suggest changing to linear regression to compare hospital cost after log-transformation. This will provide more informative and interpretable result than using quantile regression. Also, for the available sample size, bootstrap is not necessary.

For splitting patients into complication/no complication. CVD grade I patients are similar to patients with no complication, and their hospital cost will be similar to those without complication rather than CVD Grade V. Thus, grouping CDV I-II with CVD III-V together and compare them to no complication makes a strange comparison. I understand that authors want to compare complication to no complication at all, but authors should consider splitting complication groups into mild vs. severe groups for the main analysis (rather than secondary).

For the concerns in methods, results are not commented in detail.

Other problems:

LN 41-42. The abstract should provide precise and accurate result. “Almost 1 in 4” should be replaced with an accurate percentage.

LN 43-46. Should report median [IQR] cost for each and report the difference tested.

LN 164. Authors should consider grouping patients differently.

LN 167. Missing data assessment should be in the result section.

LN 182. Bonferroni correction should apply to all tested outcomes rather than just significant difference. Significance criteria should be stated clearly.

LN 185. Spearman correlation is for assessing the monotonic association, not linearity. Correlation does not make sense for binary variable (complication).

LN 188. Need to clarify criteria for variable selection.

LN 208. Authors should report total eligible patients before missing data, and missing data assessment should be reported in the result rather than method.

LN 240. Need to clarify adjusted and unadjusted hospital cost in the method section.

Supplement Table 2. Why reporting percentage across CVD groups?

Gramma. There are multiple grammar errors across the manuscript that adds difficulty to the comprehension of the content. Authors should proofread the submission carefully.

6. PLOS authors have the option to publish the peer review history of their article (what does this mean?). If published, this will include your full peer review and any attached files.

Reviewer #1: No

---

## [Author Response · Author response to Decision Letter 0]

5 Oct 2020

Prof Ehab Farag (MD FRCA FASA)

Academic Editor

PLOS ONE

3rd October 2020

KINDLY REFER TO DETAILED COVER LETTER FOR ALL IMAGES.

Re: PONE-D-20-23870: Postoperative complications and hospital costs following small bowel resections

On behalf of my co-authors I would like to sincerely thank you and the expert Reviewers for considering our revised manuscript for publication in PLOS ONE.

We are genuinely appreciative for constructive feedback and we are grateful for the opportunity to further refine the manuscript. As instructed, we provide a detailed rebuttal to each point raised by the academic editor and reviewers. Include our resubmission is:

• A marked-up copy of our manuscript that highlights changes made to the original version. This is uploaded as a separate file labeled 'Revised Manuscript with Track Changes'.

• An unmarked version of our revised paper without tracked changes. This is uploaded this as a separate file labeled 'Manuscript'.

EDITOR 1

Authors’ response: We have ensured that our manuscript conforms to the PLOS ONE style requirements. 

2. Thank you for providing information in the Ethics Statement about your ethics board approval and their waiver of the requirement for informed patient consent. However, we ask that you also provide this information within your Methods section.

Authors’ response: The Ethics Statement about our ethics board approval and their waiver of the requirement for informed patient consent is now presented in our Methods section. 

Authors’ response: Thank you for this comment. This has been completed. 

REVIEWER 1 COMMENTS

We thank Reviewer #1 for taking the time to review our submission. We are extremely grateful for the comments provided and for giving us the opportunity to further enhance and strengthen our manuscript for publication in PLOS ONE. 

Reviewer #1: I have three major concerns about the study method.

1. First, for the determination of complication. Authors needs to clarify how the complications were determined and counted, which is crucial to the validity of the study. Especially, authors need to verify that all the complication are postoperative. For example, myocardial infarction and respiratory failure both have counts under Grade I in supplement table 2?

Authors’ response: Thank you for this important comment. We strongly agree that is imperative to clarify how the complications were determined and counted. Thank you for also correctly pointing out the example of both myocardial infarction and respiratory failure being graded under a Clavien–Dindo (CVD) Grade I complication in the Supplement Table 2.

Postoperative complications were defined as any deviation from the normal postoperative course during the index admission and guided by the European Perioperative Clinical Outcome definitions. Severity of complications were graded according to the Clavien–Dindo classification system, which is a validated classification system that categorises complication severity based on the level of required treatment: CVD Grade I includes any deviation from normal postoperative course that does not require intervention, excluding antiemetics, antipyretics, analgesia, diuretics, electrolytes and physiotherapy; CVD Grade II requires pharmacological treatment, blood transfusion or total parenteral nutrition; CVD Grade III requires radiological, surgical or endoscopic intervention; CVD Grade IV includes any life-threatening complications that require intensive care management; and CVD Grade V is when death occurs. Patients were stratified into groups based on the worst complication severity recorded.

We have carefully reviewed all complications and also reviewed the original clinical medical records to ensure integrity of the data. 

Regarding the 2 cases of myocardial infarction that were originally graded as CVD grade I. A detailed review of the medical records showed that both patients fulfilled the definition of myocardial infarction according to the European Perioperative Clinical Outcome definitions. Each patient presented with an incidental finding of raised cardiac troponins with new T-wave ECG changes. Both patients had no symptoms and the diagnosis was picked up after routine cardiac troponins levels were checked by the treating unit. Cardiology advice at the time suggested a diagnosis of myocardial injury after non-cardiac surgery (MINS), and apart from prescribing aspirin, there was no further cardiac intervention. Both patients remained completely asymptomatic. We have also followed both these patients post-discharge. They have both had cardiology outpatient follow up, a negative dobutamine stress echocardiographic stress test. Because both patients were asymptomatic and the findings were incidental, they were graded as CVD grade I in our original submission. On reflection, we have revised this complication grade to a CVD grade II, given that aspirin was started for each patient. 

The workup for patients with MINS is still very controversial. Emerging evidence suggests that many patients sustain myocardial injury in the perioperative period which will not satisfy the diagnostic criteria for myocardial infarction. Myocardial injury after noncardiac surgery is common among adults undergoing noncardiac surgery and CVD grade II, as each received either supplementary oxygen therapy, oral or intravenous diuretics, or both. All other complications have been rechecked with the medical records and are correct.

2. The other concern is about the use of quantile regression model. While authors claim that hospital cost was negatively skewed (left skewed), supplement graphs showed that hospital cost are right skewed. In this case, I would suggest changing to linear regression to compare hospital cost after log-transformation. This will provide more informative and interpretable result than using quantile regression. Also, for the available sample size, bootstrap is not necessary.

3. For splitting patients into complication/no complication. CVD grade I patients are similar to patients with no complication, and their hospital cost will be similar to those without complication rather than CVD Grade V. Thus, grouping CDV I-II with CVD III-V together and compare them to no complication makes a strange comparison. I understand that authors want to compare complication to no complication at all, but authors should consider splitting complication groups into mild vs. severe groups for the main analysis (rather than secondary).

Authors’ response: Thank you for these excellent comments that have been discussed in depth amongst the authors. We have addressed these two comments simultaneously. 

As the Reviewer has pointed this data is “positively skewed” to the right side - its skewness was 6.45. In the manuscript we have now stated “Because hospital cost had a severely positive skewed distribution with a skewness of 6.45 (95%CI: 6.20 - 6.71), we used quantile regression modelling.”

As the Reviewer has suggested, we did attempt log transformation with base of 10 and natural log. Transformed total hospital cost remained skewed and the normality test (Shapiro-Wilks test) failed to prove a normal distribution of transformed data. The following Figure shows the transformed results.

Overall the performance of linear regression for three measurement methods seems appropriate. 

According to linear regression analysis, in patients with complications, hospital cost increases 53.5% (95%CI: 30.6% ‒ 80.7%) compared to patient without complications. The burden of hospital cost increases with the number and severity of complications. Patient with one complication had a 5.4% (95%CI: -12.1% ‒ 26.5%) increase in hospital cost, however patients with more than 4 complications had significantly higher costs [140.4% (95%CI: 103.2% ‒ 185.1%)] compared to patients with no complications. 

We have also shown that as the severity of complications increases, the costs increase. Hospital cost increase from 12.7% (95%CI: -5.4% ‒ 34.3%) to 235.0% (95%CI: 171.0% ‒ 313.0%) concordant with increased CVD grades, except for Grade V (representing in-hospital mortality). These patients tended to die early in postoperative period and therefore hospital cost were lower compared to patients with CVD grade IV complications who required extensive tests/interventions/ICU support and had an extended length of hospital stay. We found that patients with a CVD grade V complication incurred a 47.9% (95%CI: 15.9% ‒ 88.4%) higher cost compared to patients without any complication.

We agree with the reviewer that linear regression is an excellent statistical method to prove the relationship between several explanatory variables and a dependent variable. Its main inference process is based on a simple ordinary least square (OLS) method or weighted OLS. Consequently, the relationship is made as linear or proportional according to variable transformation or weighting. The estimated (predicted) distribution of dependent variable has to follow the distribution made by dependent variables, since their relationship is only explained by a linear regression equation. It’s important to note that hospital cost are determined by multidimensional clinical and economic factors. For example, although mortality is graded as CVD grade V (the highest severity of complication), the hospital costs for CVD grade V was lower than that of CVD grade IV. Paradoxically, as we outlined above, when patients experience CVD grade V complications in the early postoperative period, the hospital costs paradoxically decrease. 

Our findings also show that CVD grade V complications (in-hospital mortality) account for lower hospital costs than CVD grade IV complications. If we grouped cost into mild and severe groups, the data would be misleading. Accordingly, a conclusion could be made that there is a log-linear relationship between severity of complications and hospital costs, which is of course inaccurate. 

Therefore, after considered discussion, we think that quantile regression provides more granular and more accurate information of the data we have presented. Quantile regression is not limited within the traditional one representative mean value and provides several coefficients of interest. Using quantile regression, specific ranked dependent values are selected and analyzed by linear regression. 

To further demonstrate our argument, the graph below shows how quantile regression provides detailed information about hospital cost focused on the 25th, 50th and 75th quantile values.

The Figure below further demonstrates graphically the changes in quantile coefficients along with 95%CI. The first figure presents the coefficient changes along with the percentile distribution of hospital cost from the quantile regression, which estimated with the presence of complications as a dependent variable. Red colored lines represent the coefficient and 95%CI estimated by the ordinary least square method. 

The quantile regression coefficients throughout the range were confined mostly within 95% CI of the OLS coefficient, which implies that the estimated coefficients by two method could be similar. However, coefficient estimated in the high percentile cost range tends to increase even confined within 95%CI (red colored dashed lines) by OLS coefficient. This trend appears apparently in higher number and severity of complications, especially in the patients with more than 4 complications and CVD grade IV and V. 

Quantile regression coefficients plot: Hospital cost vs. presence of complications

Quantile regression coefficients plot: Hospital cost vs. number of complications

Quantile regression coefficients plot: Hospital cost vs. severity of complications

Quantile regression estimators can be computed easily using a linear modelling with proposed characteristics by Koenker. The distribution of variance of quantile regression estimators are generally unknown because they are based on their rank. One reliable method to solve this problem of quantile regression is “resampling methods; bootstrapping” and we applied simple bootstrapping methods for quantile regression with the number of 200 replications recommended as Bilias (2000).

In conclusion, we think that quantile regression delivers more granular and informative results compared to linear regression, especially when trying to accurately understand the characteristics of complications as an important cost driver. 

In the revised manuscript we have provided a detailed description explaining our rationale for quantile regression modelling. We state “Because hospital cost had a severely positively skewed distribution (skewness of 6.45: 95% CI: 6.20–6.71), we used quantile regression modeling to investigate the cost-driving effects of complications according to low (25th quantile), median (50th quantile), and high (75th quantile) cost brackets. Spearman’s correlation analysis was performed to clarify which variables were in a relationship with complications and hospital costs. Based on the correlation analysis results (see Supplementary Figure 1) and considering the clinical relevance, several variables were then selected for the adjusted regression analysis”.

References

Koenker R. Quantile regression for longitudinal data. Journal of Multivariate Analysis 91.1 (2004): 74-89.

Koenker R, et al., eds. Handbook of quantile regression. CRC press, 2017.

Koenker R. Quantile Regression. Cambridge University Press, Cambridge, 2005.

Wenz, Sebastian E. "What quantile regression does and doesn't do: A commentary on Petscher and Logan (2014)." Child development 90.4 (2019): 1442-1452.

Bilias, Y. Chen, S. and Z. Ying, (2000) Simple resampling methods for censored quantile regression, J. of Econometrics, 99, 373-386.

Other problems:

5. Line 41-42. The abstract should provide precise and accurate result. “Almost 1 in 4” should be replaced with an accurate percentage.

Authors’ response: Thank you for this important comment. Thank you for this comment. We provided the accurate percentile for the patients with CVD grade III or IV. 

We now state “The overall complication prevalence was 81.6% (95% CI: 85.7–77.5). Most complications (69%) were minor, but 22.9% of patients developed a severe complication (Clavien–Dindo grades III or IV).”

6. Line 43-46. Should report median [IQR] cost for each and report the difference tested.

Authors’ response: Thank you for pointing out our error. We apologize for reporting the incorrect numbers, and have corrected appropriate numbers with interquartile range: “(median [IQR] USD 19,659.64 [13,545.81 – 35,407.14] vs 11,551.88 [8,849.46 – 15,329.87], P<0.001)”

7. Line 164. Authors should consider grouping patients differently.

Authors’ response: Thank you for this comment. We hope we have addressed this important point in our comments above. 

8. Line 167. Missing data assessment should be in the result section.

Authors’ response: Thank you for this excellent suggestion. We have inserted the following information in methods section “Before statistical analysis, missing data analysis was performed to detect more than 5% missing values for all variables. For variables with less than 5% of missing values, statistical analysis excluding cases by analysis was planned. The multiple imputation method was performed in cases of missing values of more than 5%”.

Further we have moved the missing data analysis into the Results section with the number of eligible patients. We now state “Among the data of 348 patients, missing data analysis demonstrated fewer than 5% missing values for all variables. The variables with the highest missing data rate were ‘preoperative bilirubin concentration’ (4.3%), ‘intraoperative crystalloid administration volume’ (3.4%), ‘preoperative albumin concentration’ (2.9%), and ‘lowest postoperative hemoglobin concentration’ (0.9%). Statistical analysis was performed as a complete case analysis.”

9. Line 182. Bonferroni correction should apply to all tested outcomes rather than just significant difference. Significance criteria should be stated clearly.

Authors’ response: Thank you for raising this very important point. We agree with the Reviewer’s and have now applied the Bonferroni correction to the quantile regression analysis. We measured postoperative complication using three measuring scales: presence (dichromatic), numbers (5 levels ordered categorical), CVD grade (6 level ordered categorical). Accordingly, we have re-evaluated our results by the guide of Bonferroni’s adjusted P value = 0.050/3 = 0.016. Remaining conservative manner, we have limited the P value to 0.016, not as 0.017. We have described this new significance limit in statistical analysis section. 

We now state “To correct for multiple comparisons the Bonferroni correction was applied.”

Throughout the manuscript, our statistical results have all been revised to take into consideration the new adjusted P value limit.

10. Line 185. Spearman correlation is for assessing the monotonic association, not linearity. Correlation does not make sense for binary variable (complication).

Authors’ response: We absolutely agree with the Reviewer and thank the Reviewer for pointing out our error. All relevant sentences related to correlation analysis have now been corrected.

11. Line 188. Need to clarify criteria for variable selection.

Authors’ response: Thank you for this insightful comment. We have included a detailed explanation about why we chose the specific variables for adjusting regression modelling. 

Authors’ response: In the revised manuscript we state “Supplementary Figure 1 presents the Spearman correlation analysis results among complications, total hospital cost, and other collected variables. Among the collected variables, the following were selected as covariates: the CCI, preoperative anemia, surgical technique, emergency surgery, the volume of intraoperative fluid administration, and transfusion during admission. The CCI is a representative variable for preoperative patients’ status. Preoperative anemia significantly correlated with complications and hospital costs; it was selected as a covariate because of its important clinical relevance. Although its correlation coefficient was weak, the surgical technique consistently correlated with complications and hospital costs.

Emergency surgery was selected as a covariate because it is not included in the CCI classification. Intraoperative fluid volume had a weak relationship with the number of complications and hospital costs; however, as it can be related to postoperative complications, it was selected as a covariate (16). Transfusion during admission showed a moderate relationship with the severity of complications and hospital costs, and a weak relationship with the number of complications. As transfusion and anemia are associated with the development of postoperative complications, they were selected as covariates (17, 18). Although age and operative time correlated with complications and hospital costs, they were excluded because of multicollinearity. Other variables were excluded due to a non-significant correlation or clinical irrelevance.”

We have added 3 additional references to support our statement.

12. Line 208. Authors should report total eligible patients before missing data, and missing data assessment should be reported in the result rather than method.

Authors’ response: Thank you for raising this important point which has now been addressed in the resubmission manuscript. We have also provided a more detailed response to this important comment in the response above.

13. Line 240. Need to clarify adjusted and unadjusted hospital cost in the method 

Authors’ response: Thank you for this valuable comment. We have now introduced a detailed explanation about unadjusted and adjusted cost analysis in the method section under the heading statistical analysis. We now state “Total hospital costs in relation to complications were analyzed using unadjusted and adjusted hospital costs. For the adjusted analyses, costs were analyzed according to the occurrence, number, and severity of complications using covariates of both clinical and statistical importance.”

We have also re-arranged the sentences to enhance the presentation of this additional information.

14. Supplement Table 2. Why reporting percentage across CVD groups?

Authors’ response: We agree with the Reviewer’s comment and have removed all the percentages across the CVD groups. Thank you for this excellent suggestion. We hope that the revised Table is easier to read, more informative and less confusing. 

15. There are multiple grammar errors across the manuscript that adds difficulty to the comprehension of the content. Authors should proofread the submission carefully.

Authors’ response: Thank you for this comment. The manuscript has undergone a professional edit to ensure that the grammar, syntax and sentence structure is exemplary. All changes have been highlighted as a Tracked change in red.

Once again, we would like to the Editors and expert Reviewers for considering our manuscript for publication in PLOS ONE. The constructive comments and expert advice is genuinely appreciated. 

Best, 

A/Prof Laurence Weinberg

(BSc, MBBCh,MRCP,DPCritCareEcho,FANZCA,MD)

Chairperson, Austin Health Human Research Ethics Committee; 

Director, Department of Anesthesia, Austin Hospital

Associate Professor, Department of Surgery, Austin Health, University of Melbourne

Associate Professor, Perioperative Pain and Medicine Unit, Department of Surgery, University of Melbourne

---

## [Editor Report · Decision Letter 1]

7 Oct 2020

Postoperative complications and hospital costs following small bowel resection surgery

PONE-D-20-23870R1

Dear Dr. Laurence Weinberg

We’re pleased to inform you that your manuscript has been judged scientifically suitable for publication and will be formally accepted for publication once it meets all outstanding technical requirements.

Kind regards,

Ehab Farag, MD FRCA FASA

Academic Editor

PLOS ONE
---

## [Editor Report · Acceptance letter]

9 Oct 2020

PONE-D-20-23870R1 

Postoperative complications and hospital costs following small bowel resection surgery 

Dear Dr. Weinberg:

I'm pleased to inform you that your manuscript has been deemed suitable for publication in PLOS ONE. Congratulations! Your manuscript is now with our production department. 

Kind regards, 

on behalf of

Dr. Ehab Farag 

Academic Editor

PLOS ONE